# Purification and Glutaraldehyde Activation Study on HCl-Doped PVA–PANI Copolymers with Different Aniline Concentrations

**DOI:** 10.3390/molecules24010063

**Published:** 2018-12-25

**Authors:** Jorge M. Guerrero, Amanda Carrillo, María L. Mota, Roberto C. Ambrosio, Francisco S. Aguirre

**Affiliations:** 1Centro de Investigación en Materiales Avanzados, S.C., Alianza Norte 202, Parque de Investigación e Innovación Tecnológica, Apodaca, NL C.P. 66600, Mexico; jorge.guerrero@cimav.edu.mx; 2Instituto de Ingeniería y Tecnología, Universidad Autónoma de Ciudad Juárez, Av. Del Charro 610, Ciudad Juárez, CHIH C.P. 32310, Mexico; mdllmotago@conacyt.mx; 3CONACYT, Universidad Autónoma de Ciudad Juárez, Ciudad Juárez, CHIH C.P. 32310, Mexico; 4Facultad de Electrónica, Benemérita Universidad Autónoma de Puebla, Puebla C.P 72000, Mexico; roberto.ambrosio@correo.buap.mx

**Keywords:** polymerization dispersion method, polyaniline, polyvinyl alcohol, glutaraldehyde, chemical activation

## Abstract

In this work, we report the synthesis and purification of polyvinyl alcohol-polyaniline (PVA–PANI) copolymers at different aniline concentrations, and their molecular (^1^H-NMR and FTIR), thermal (TGA/DTG/DSC), optical (UV–Vis-NIR), and microstructural (XRD and SEM) properties before and after activation with glutaraldehyde (GA) in order to obtain an active membrane. The PVA–PANI copolymers were synthesized by chemical oxidation of aniline using ammonium persulfate (APS) in an acidified (HCl) polyvinyl alcohol matrix. The obtained copolymers were purified by dialysis and the precipitation–redispersion method in order to eliminate undesired products and compare changes due to purification. PVA–PANI products were analyzed as gels, colloidal dispersions, and thin films. ^1^H-NMR confirmed the molecular structure of PVA–PANI as the proposed skeletal formula, and FTIR of the obtained purified gels showed the characteristic functional groups of PVA gels with PANI nanoparticles. After exposing the material to a GA solution, the presence of the FTIR absorption bands at 1595 cm^−1^, 1650 cm^−1^, and 1717 cm^−1^ confirmed the activation of the material. FTIR and UV–Vis-NIR characterization showed an increase of the benzenoid section of PANI with GA exposure, which can be interpreted as a reduction of the polymer with the time of activation and concentration of the solution.

## 1. Introduction

Conductive polymers have been studied since their discovery in 1977 [1]. Among all known conductive polymers, polyaniline (PANI) has been one of the most studied due its high environmental stability, straightforward control of its chemical and physical properties through doping, and relatively low cost of development in comparison to other conductive polymers. PANI has been the subject in numerous studies for applications such as membranes [2], anticorrosive coatings [3], biosensors [4,5,6,7], and electronic devices [8,9]. However, the implementation of PANI has been limited because of its poor mechanical properties and its poor solubility in most organic solvents. This insolubility results in heterogeneous solutions, where the presence of microparticles hinders the formation of homogeneous PANI thin films at low cost by physical methods such as spin or dip coating. To overcome these drawbacks, several methods have been studied, such as the possibility of processing PANI in the form of mixtures with electrically insulating polymers, improving the presented deficiencies, and opening a range of potential applications. In recent years, several publications report on the use of polystyrene (PS) [10], polyvinyl chloride (PVC) [11], and polyvinyl alcohol (PVA) [12], to stabilize PANI nanoparticles and improve the processing and formation of thin films. 

The preparation of PVA–PANI copolymers by polymerization dispersion has already been reported showing a stable colloidal dispersion without sedimentation [13]. Also, with the increase of PVA content the stability of the colloidal dispersion increases, improving its mechanical properties [14]. Nevertheless, PVA addition affects the conductivity of the material. Work has been done to correlate the mechanical and electrical properties of PVA–PANI as function of aniline concentration [15].

Both PANI and PVA–PANI have been activated with glutaraldehyde (GA) to obtain PANI-G and PVA–PANI-G materials that can be used as biological platforms for the detection of different analytes such as enzymes and proteins [16,17]. The biocompatibility of PVA is already known, but PVA–PANI obtained by polymerization dispersion requires further purification in order to be applied in the biomedical field because of the presence of undesired byproducts produced by the chemical polymerization of aniline [18], which can also affect the activation of the material. 

The present work is focused on the study of purified PVA–PANI copolymers at different concentrations of aniline. The molecular, thermal, optical, and microstructural properties before and after PVA–PANI copolymer activation with GA have been studied. The aim of this work is to provide information about the effects of purified PANI in a PVA matrix with GA activation.

## 2. Results and Discussion

### 2.1. Molecular Structure Variation and Modification of PVA–PANI Copolymers 

#### 2.1.1. Proton Nuclear Magnetic Resonance Spectroscopy (^1^H-NMR)

To obtain the molecular structure of PVA–PANI copolymers, ^1^H-NMR analysis (500 MHz, TMS) was carried out using deuterium oxide (D_2_O) as solvent. The obtained spectrum for PVA–PANI is shown in Figure 1 where at δ = 7.3 ppm and δ = 7.5 ppm there is evidence of a doublet of doublets (d, *J* = 6.5 Hz, C_6_H_4_) corresponding to the benzenoid and quinoid sections of PANI in emeraldine phase, then δ = 4.7 ppm (s, D_2_O). The doublet located at δ = 3.9 ppm (d, *J* = 28.5 Hz) corresponds to the characteristic signal of α proton of oxygen binding PVA–PANI, while the multiplet located at δ = 1.5 ppm (m, *J* = 18.7 Hz) suggests the integration of methylene protons from the PVA backbone [19,20].

The smooth baseline of the spectrum suggests absence of impurities in the obtained material (Figure 2). After water suppression the actual PVA:PANI ratio can be calculated by integrating the signal corresponding to PVA and PANI, obtaining a 2:1 ratio (PVA:PANI). Moreover, two sharp singlets can be clearly seen at δ = 2 ppm and δ = 3.25 ppm corresponding to absorbed solvent in the copolymer. The δ = 4.7 ppm (d, *J* = 19 Hz), can correspond to the proton of PVA hydroxyl (–OH), pendant groups or PANI secondary amine (–NH) groups within the polymer chain. The doublet located at δ = 3.9 ppm (d, *J* = 28.5 Hz) corresponds to the characteristic signal of α proton of oxygen binding PVA–PANI, this completes the molecular structure of the PVA–PANI copolymers as proposed from the skeletal formula in Figure 2. Is important to mention that the signals could be clearly seen only after the water suppression and is considered to be masked by the solvent peak in Figure 1.

#### 2.1.2. Fourier Transform Infrared Spectroscopy (FTIR)

To confirm the structure of PVA–PANI and evaluate chemical changes upon activation with GA, an FTIR analysis was carried out. First, a comparison of both purification methods was made through a study of the functional groups present (Figure 3). It was observed that PVA–PANI purified by the precipitation–redispersion method presents the characteristic bands as follows, O–H between 3600 and 3000 cm^−1^, aliphatic groups for the PVA backbone between 2900 and 2800 cm^−1^, and stretching of C=O and C–O acetate groups from hydrolyzed PVA in the fingerprint region [21]. The presence of the bands at 1179, 1031, 739, and 689 cm^−1^ corresponding to C–N and C–H vibrations from the benzenoid and quinoid section of PANI confirm the presence of the polymer in the PVA matrix. On the other hand, the spectrum for the dialysis purified material shows the characteristic bands for PVA–PANI at a medium–high concentration of aniline in a PVA matrix, according to previous studies [15]. The N-H, C=C, C–C, and C–N vibrations centered at 3383, 1600, 1423, and 1297 cm^−1^, respectively, can be seen along with the C–H in plane and out of plane vibrations from the quinoid and benzenoid rings [22]. Also, from Figure 3, it is seen that both purification methods do not show any uncoupled or overlapped bands by the content of unreacted material, especially in the fingerprint region of the spectrum. 

The implementation of dialysis has been used for the purification of PANI, since it provides a better dispersion of particles and minimizes product loss. However, filtration of the product obtained from precipitation–redispersion is highly recommended, since it takes a short time and allows the obtaining of only the desired copolymer in gel form through precipitation. Figure 4 shows the corresponding bands of PVA gels modified with PANI (ES) nanoparticles obtained by the precipitation–redispersion method with different concentrations of aniline (Table 1). An increase in the absorption bands with the increase of monomer in the PVA matrix was observed.

The absorption bands located between 3600 and 3000 cm^−1^, attributed to O–H and N–H groups of PVA and PANI, respectively, show an intensity increase which is due to the functional groups in this range and widening due to minimum wavenumber displacements. Subsequently, in the range of 2350 to 1750 cm^−1^, small absorption bands can be observed corresponding to the presence of overtones from PVA–PANI. These bands show an intensity decrease with increasing aniline in the polymer matrix. The latter can be attributed to chain length variation of the obtained PANI for each experiment. An increase in the absorption band located at 1586 cm^−1^ indicates an increase in C=C bonds associated to quinoid rings which can also be correlated to significant production of PANI in the PVA matrix. The absorption bands found at 1416, 1369, and 1320 cm^−1^ correspond to the absorption of C–C stretching of the quinoid ring in PANI, the flexion of the C–H group, and the stretching vibration of the C–N bond of the amine in the aromatic ring, respectively. Absorption bands due to the vibration of the C–N bond and the stretching of the C–H group from the aromatic ring within the plane are shown at 1230 and 1130 cm^−1^, respectively. The band located at 1090 cm^−1^ corresponds to the stretching vibration of the C–O bond which tends to be much higher in PVA–PANI composites, showing a more pronounced shoulder in 1031 cm^−1^ with the increase of monomer, demonstrating a cross-linking between the materials [23]. The FTIR spectra obtained for the activated material with 1% GA solution for an activation time of 5, 15, and 30 min can be observed in Figure 5a–c, respectively.

In Figure 5, a reduction in the intensity of the absorption bands corresponding to the O–H and N–H groups of PVA–PANI is shown, indicating an interaction between the activating solution and the polymer [24]. The inset shows the range from 1550 to 1750 cm^−1^, where the presence of three absorption bands can be observed at 1595, 1650, and 1717 cm^−1^, corresponding to the presence of C=N (Schiff base) bonds formed by the linkage of primary amines with C=O groups, and the presence of free C=O groups [24,25]. As the activation time so does the absorption band at 1595 cm^−1^, correlated to the vibration of the C=N (imine) bond, overlapping the vibration of the C=C bond of the quinoid section of PANI. Moreover, an increase in the absorption band located at 1490 cm^−1^ can be observed, which is associated to the benzenoid ring stretching [22], along with a reduction of the intensity of the C–C stretching from the quinoid section, which can indicate a reduction of the purified material with the increase of activation time [26]. Further activation of PVA–PANI with a more concentrated solution (2.5% GA) confirmed the reduction of the material as well as a branching between both materials (Appendix A).

#### 2.1.3. UV–Visible-Near-Infrared Spectroscopy (UV–Vis-NIR)

UV–Vis-NIR absorption spectra and its normalized form for PVA–PANI films at low, medium, and high concentrations of PANI can be observed in Figure 6. Absorption bands corresponding to π–$\pi^{*}$ transitions from the excitation of benzenoid segments (~330 nm), polaron-$\pi^{*}$ transitions associated with benzenoid and quinoid ring (~430 nm), and π-polaron transitions from the excitation of quinoid rings from doped PANI (~800 nm) can be seen in each experiment [3,27]. The absorption bands corresponding to π–$\pi^{*}$ transitions show a bathochromic effect as a function of the increase in aniline concentration in the PVA matrix, attributed to an increase in the molecular weight of the produced PANI [28]. Furthermore, a clear hyperchromic effect is observed, associated to the increase in concentration of the material, as established by Beer–Lambert’s law.

According to the results of activated PVA–PANI obtained by FTIR, an immersion time of 30 min was used for the PVA–PANI-G films. The UV–Vis-NIR absorption spectra and its normalized form for PVA–PANI-G films at low, medium, and high aniline concentrations are shown in Figure 7.

It can be seen that the absorption bands of the obtained PVA–PANI-G films present an hypsochromic effect for low and medium aniline concentrations, and a bathochromic effect for medium and high aniline concentrations, which can be attributed to a change in polarity of the molecule when interacting with the aldehyde groups of GA. The characteristic absorption bands for the electronic transitions of PANI, as well as the characteristic shift due to the increase in aniline concentration, according to the PVA–PANI experiment shown in Table 1, are also seen in Figure 7. A hyperchromic effect can be seen in the absorption band located at 380 nm, which can be related to an increase of the benzenoid section of PANI due to the activation time and concentration of GA. Comparison of bandgap variations due to concentration and activation of the material are shown in Appendix A.

### 2.2. Purity, Thermal Stability and Cross-Linking of PVA–PANI Copolymers

Thermogravimetric Analysis, Derivative Thermogravimetric Analysis, and Differential Scanning Calorimetry (TGA/DTG/DSC)

In order to evaluate the thermal stability, the purity of the obtained materials by both purification methods, and to correlate an increase in hydroxyl and amine functional groups, TGA-DTG analysis were carried out. DSC study was performed to observe the cross-linking effects on PVA–PANI and PVA–PANI-G in order to evaluate the sensitivity of the material towards GA. First, to confirm purification of the material, an analysis of the transition temperatures and molecular weight for the obtained byproducts and sideproducts of the synthesis route for PVA–PANI (Figure 8) was attained as shown in Figure 9.

We expected to observe a weight loss for each undesired product along with an exothermic peak in the temperature range of the thermograms presented in Figure 10. TGA and DTG of the material purified by precipitation–redispersion method (Figure 10a) show a loss of 8.41 wt % between 25 to 172 °C, which can be attributed to solvent and moisture loss at the surface of the material. The absence of an exothermic peak in this range and a gradual weight loss can be related to the absence of undesired products that can present a transition in this range of temperature, such as APS and HCl. Subsequently, the presence of two exothermic peaks located at 173.9 and 193.3 °C is associated to the removal of bound water, moisture, and dopant molecules inside the polymer matrix, as well as the loss of free hydroxyl functional groups due to the decomposition of PVA [29] (mass loss of 21.97 wt %). At ~250 °C an exothermic peak is shown as a shoulder corresponding to the breaking of free amine functional groups associated to the decomposition of the PANI polymeric chain, which continues to gradually decrease along with the dehydroxylation of the copolymer up to 395 °C (total mass loss of 44 wt %). In the range of 400 to 500 °C several peaks associated with the degradation of similar materials can be observed, which can be related to a variation in distribution of different molecular weights of PANI. On the other hand, PVA–PANI purified by dialysis (Figure 10b) shows a weight loss of 12.37%, which is attributed to moisture and the removal of HCl from the material [12], showing an exothermic peak at 52.94 °C. In the same way as the material purified by precipitation–redispersion, the decomposition of PVA and PANI is shown at 199.55 °C and 250.74 °C, respectively, to which the gradual loss of hydroxyl and amine functional groups can be related, having a total weight loss of 20.6% of the material. The presence of a small exothermic band at 355.25 °C can be attributed to a cross-linking reaction in the polymer matrix and not to the presence of oligomers in the material, which can be confirmed with the presence of a single exothermic peak at 489.15 °C associated with the degradation of the copolymer with a loss of 16.95 wt %. Also, from Figure 10, it can be seen that PVA–PANI purified by dialysis shows improved thermal stability and molecular integration which is evidenced by the presence of peaks correlated to decomposition and degradation of PVA and PANI only; this can be attributed to the processing of the material during the purification process.

After analyzing the thermal stability and purity of the material obtained by both purification methods, the effect of aniline concentration was studied. Figure 11 shows three weight losses. The first weight loss in the temperature range from 25 to 160 °C can be attributed to the loss of moisture and free acid, corresponding to a weight loss of 9.76%. There is another weight loss between 160 and 400 °C which is attributed to degradation of short polymer chains along with the removal of hydrogen bonds and Coulomb interactions from O–H and N–H functional groups, corresponding to a weight loss of 40.24% [30,31]. With regard to the increase of aniline, a shift to higher temperatures of 21.14 °C can be observed starting from a low concentration level to a high concentration of PANI from room temperature to 400 °C, which corresponds to a weight loss of 27%, 29%, and 30% for low, medium, and high concentrations, respectively, indicating that an increase in molecular weight of PANI contributes to a higher thermal stability, but also a mayor weight loss because of the presence of more hydroxyl and amine functional groups in the polymer chain. The decomposition of the purified PVA–PANI gels tends to be gradual until 400 °C where there is another pronounced weight loss of 36.3% due to the degradation of the material, leaving a residue of 14.5 wt % with a variation of 1.21% between each concentration.

Correlation of previously obtained thermal stability and cross-linking was studied by DSC, in order to observe energy variations associated to interactions of the polymer matrix with the PANI nanoparticles. According to the obtained DSC results (Figure 12), the presence of an endothermic peak at ~200 °C can be attributed to a cross-linking reaction between PVA and PANI at that temperature, where enthalpy changes are inversely proportional to the concentration of chemically linked PANI [32]; indicating that PVA–PANI at low concentration of monomer shows a better cross-linking between the materials. Also, it can be seen that no other prominent transition peaks are shown in the thermogram, indicating that the purified material is amorphous.

The obtained TGA-DTG results for PVA–PANI-G gels are shown in Figure 13. First, a weight loss of 6 to 9% can be observed, which corresponds to the loss of absorbed solvent and moisture. Subsequently there is a weight loss of 33 to 41% corresponding to the loss of pendant functional groups and hydroxyl and/or amino terminal groups. Starting at 400 °C, the degradation of the copolymer is seen, which no longer presents a gradual and constant decomposition as shown in Figure 11. At this range of temperature, the material shows two stages of weight loss, one with an almost linear loss with respect to the increase in temperature and another with gradual degradation, which can be attributed to a modification in the main chain of the polymer caused by the interaction with GA. PVA–PANI at high concentration activated for 30 min showed the best thermal properties.

The DSC analysis for the previous gels is shown in Figure 14. Comparing the thermogram from Figure 12 to the obtained results for the activated copolymers, it is seen that a decrease in transition temperatures is obtained for low aniline concentration along with a decrease in enthalpy from −285.95 J/g to a minimum of −320.86 J/g at 15 min of activation, which indicates that cross-linking is favored with an increase of GA for this particular concentration. However, with an increase in activation time, the material shows an increase of enthalpy up to a maximum of −56.69 J/g for a high concentration of PANI at 30 min of activation, meaning that cross-linking between PVA–PANI and GA is disfavored, which can be related to a molecular disarrangement produced by branching of PVA–PANI, making it difficult to cross-link. Results for a higher concentration of GA are provided in Appendix A.

### 2.3. Microstructural Variation of PVA–PANI Copolymers 

#### 2.3.1. X-ray Diffraction (XRD)

XRD analysis was carried out to observe if the material crystallinity is affected by the purification process or the concentration of PANI nanoparticles in the PVA matrix. In Figure 15, it is observed that both purification processes seem to reduce the crystallinity of the thin films showing a broad peak between 16 and 40° (2θ), where PVA–PANI thin films purified by dialysis tend to show a better molecular order than PVA–PANI purified by the precipitation–redispersion method, presenting a diffraction peak located at 2θ = 19.77° which corresponds to the (101) plane of semicrystalline PVA with d_101_ = 4.485 Å [29,33]. These results were compared to the XRD results obtained from unpurified PVA–PANI thin films with high PVA concentration, showing the discussed diffraction peak without any trace of PANI peaks at 2θ ≈ 25°, which indicates that the obtained PANI is amorphous.

From the data shown in Figure 16a, it can be observed that the purified PVA–PANI doped with HCl 1 M (pH = 1) has a low atomic order regardless of aniline concentration, showing a broad peak with a maximum intensity found at 2θ = 23.29°, which is in agreement with the reported for PANI doped with HCl [34,35]. It is expected to have a lower crystallinity of the copolymer with GA activation due to the disarrangement produced by cross-linking, increasing molecular dispersion, and broadening of the XRD peaks [36]. This behavior can be observed for PVA–PANI-G thin films at low, medium, and high concentrations of aniline (Figure 16b).

#### 2.3.2. Scanning Electron Microscopy (SEM)

Figure 17 shows the SEM micrographs obtained for PVA–PANI films purified by the precipitation–redispersion method for low, medium, and high aniline concentrations with respect to the PVA polymer matrix. Similar results were obtained for PVA–PANI-G thin films (Appendix A).

The morphology of the particles presents as needles which changed with increasing aniline concentration, generating particles in granular form which tend to agglomerate in different nucleation points, according to the reports of Gangopadhyay et al. [12]. On the other hand, the morphology observed for both PVA–PANI and PVA–PANI-G purified by dialysis maintained a spherical shape with an average size of 156 nm (Figure 18), which has been reported for various PANI synthesis processes [37]. This indicates that the precipitation–redispersion process (Figure 17) influences morphological changes in the material, along with pH, temperature, and solvents used.

Even though a significant difference in morphology is shown between both purification methods, we found no significant difference in the crystallinity of the material according to Figure 15, where PVA–PANI purified by dialysis tends to show a better molecular order due to the stabilizing effect of the PVA, promoting better control in the growth of nucleation sites for the obtained films.

## 3. Materials and Methods 

### 3.1. Materials

The reagents used for the synthesis of PVA–PANI copolymers are as follows. Polyvinyl alcohol (PVA) (130,000 MW, 99% hydrolyzed), high purity aniline monomer (≥99.5%), and ammonium persulfate (APS) (ACS ≥ 98%) (Sigma Aldrich Co., Toluca, Edo. Mex., Mexico). Hydrochloric acid (36–38%) (J.T. Baker, Phillipsburg, NJ, USA) and glutaraldehyde solution (50 wt % in H_2_O) (Sigma Aldrich Co.) for the activation of the obtained PVA–PANI blends. Aniline was stored in a dark environment and under refrigeration, all reagents were used as acquired.

### 3.2. Synthesis of PVA–PANI

First a solution of PVA 5 wt % was made by dissolving 0.0202 mmol (2.6315 g) of PVA (MW = 130,000 g/mol) in 50 mL of deionized water (Milli-Q) under constant magnetic stirring at 80 °C until a clear solution was obtained. Then, the PVA 5 wt % was used as a polymer matrix for the polymerization of aniline at different ratios with respect to the PVA as shown in Table 1.

Subsequently, the pH of the matrix was adjusted to a value of ≤2, and it was taken immediately to an ice bath (T ≤ 5 °C) where ammonium persulfate was added drop wise at a ratio of 1:1 M to aniline according to Table 2. After a few minutes, the color change of the colloidal dispersion was observed, obtaining a dark green dispersion without sedimentation as reported in the literature [13].

### 3.3. Purification Process

When carrying out the PVA–PANI synthesis by the polymerization dispersion method, a number of byproducts related to doping, oxidant material (APS) and oligomers are generated (Figure 8). Therefore, these byproducts must be eliminated by means of a purification process, in order to obtain the PVA–PANI material without any other product that influences the effects produced by the material in future characterizations.

The purification of PVA–PANI copolymers was carried out mainly by two methods: precipitation–redispersion based on a solvent system CH_3_OH:H_2_O (5:1) in which the copolymer precipitates, then this precipitate is filtered, dried in vacuum, and redispersed in deionized water (Milli-Q) under temperature and continuous agitation. As well as a dialysis process, which is based on the implementation of membranes with a cut out molecular weight (MWCO) of 12,000 Da (Obtained from Sigma Aldrich Co.), where a certain amount of PVA–PANI is placed against deionized water (Milli-Q) for 48 h so that most byproducts permeate through the membrane. Prior to its chemical and thermal characterization, the obtained PVA–PANI was vacuum-filtered and dried.

### 3.4. Thin Film Development

PVA–PANI thin films were obtained by dip coating (for precipitation–redispersion purification) and spin coating (for dialysis purification) physical methods using Corning glass substrates previously cleaned with acetone, isopropanol, and deionized water (Milli-Q). Once the thin films were obtained they were washed, dried with N_2_ gas, and kept in a desiccator before being analyzed.

### 3.5. Activation of PVA–PANI Thin Films

Activation of PVA–PANI films with GA was carried out by immersing them in a 1% dilution of GA for a time of 5, 15, and 30 min at room temperature. Followed by a wash with deionized water (Milli-Q) to remove any agent that did not reacted. Later they were placed in a desiccator for storage.

### 3.6. Characterization Methods

Purity and molecular structure of the obtained PVA–PANI blend was confirmed by ^1^H-NMR using a Brucker NMR equipment (Brucker, Billerica, MA., USA) at a frequency of 500 MHz, using tetramethyl silane (TMS) as standard at room temperature. To complement molecular structure and observe the present functional groups in the material an FTIR analysis was carried out using a Thermo Scientific Nicolet iS10 FT-IR spectrometer (ATR) in the region from 400 to 4000 cm^−1^ in air atmosphere and at room temperature. Optical properties and transitions of the developed PVA–PANI thin films were analyzed using a Jenway 6850 UV–Vis-NIR spectrophotometer (Bibby Scientific, Staffs., UK) in the region from 1000 to 300 nm in absorption mode. The thermal stability, composition, purity, and cross-linking effect of the PVA–PANI gels were analyzed using a TGA/DSC SDT-Q600 implementing a temperature program from 25 °C to 600 °C with a heating speed of 10 °C/min and controlled N_2_ atmosphere. Microstructural characteristics for the thin films were obtained by XRD using a Panalytical diffractometer with a Cu kα radiation with a 1.54 wavelength used at a glancing angle. And the morphological variations of the material were observed with a Nova NanoSEM 200 equipment (FEI, Tokyo, Japan) using secondary electrons.

## 4. Conclusions

The purification of HCl-doped PVA–PANI copolymers at different aniline concentrations and the effect of their activation with GA was studied. ^1^H-NMR and FTIR analysis was used to confirm the molecular structure of the material as proposed from the skeletal formula before carrying out further characterizations. Regarding the activation of the material, it was observed that GA has a reduction effect on the PVA–PANI copolymer (increasing the number of benzene units of PANI), which increases proportionally with immersion time and concentration as confirmed by FTIR and UV–Vis-NIR. The activation of the material was confirmed by FTIR showing the characteristic absorption bands in the 1550 to 1750 cm^−1^ range, where the C=N (Schiff base) bond corresponding to the formation of imines can be found. Further analysis of cross-linking between PVA–PANI and GA was carried out by DSC, showing that the interaction of GA towards the material is favored at low aniline concentration, where minor variations where observed due to doping level. The purity of the material was confirmed by TGA-DTG, which showed bands corresponding to weight losses and exothermic peaks of PVA–PANI only. Moreover, it was seen that the precipitation–redispersion method seems to affect the structural and morphological properties of the PVA–PANI at different aniline concentrations, even after GA activation, as shown from the XRD and SEM results. Therefore, the dialysis purification method is recommended for upcoming experiments. We propose, as future work, to carry out experiments immobilizing an antibody on the activated material to test it as a platform for biological detection, in order to apply it as an active membrane in future in vitro diagnostic devices.

## Figures and Tables

**Figure 1 molecules-24-00063-f001:**
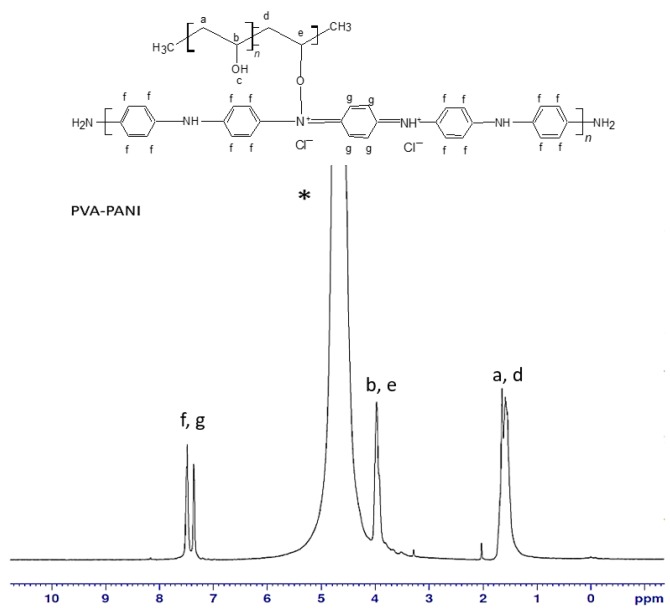
^1^H-NMR spectra for PVA–PANI in D_2_O (*).

**Figure 2 molecules-24-00063-f002:**
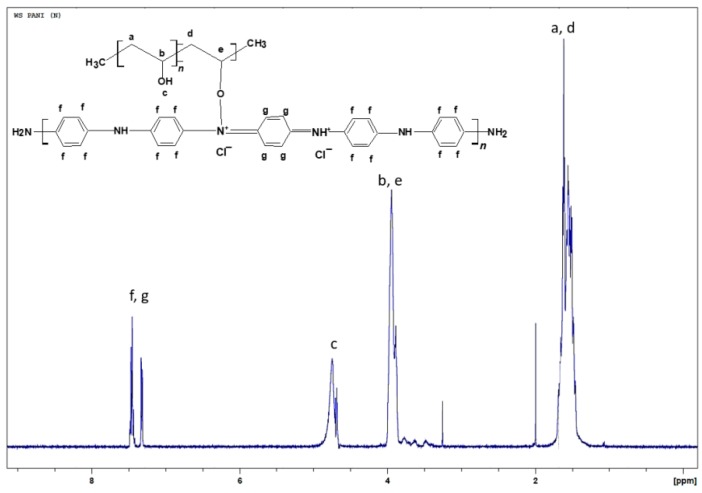
^1^H-NMR spectra for PVA–PANI with water suppression.

**Figure 3 molecules-24-00063-f003:**
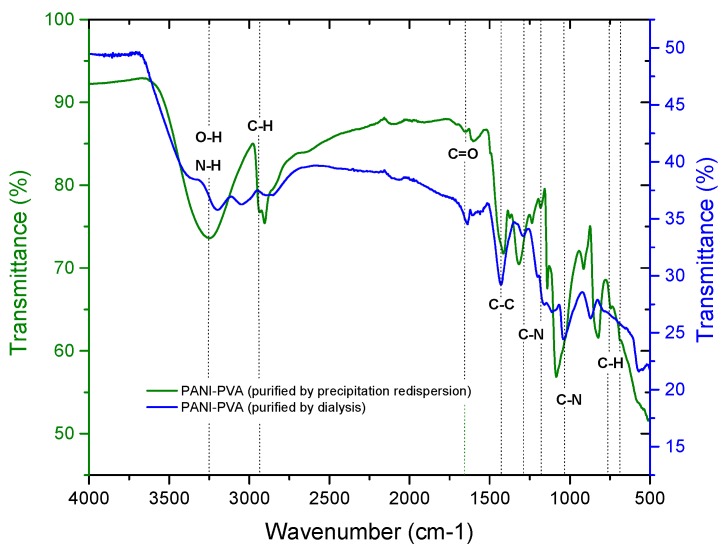
FTIR spectra of PVA–PANI purified by dialysis (-) and by precipitation–redispersion (-).

**Figure 4 molecules-24-00063-f004:**
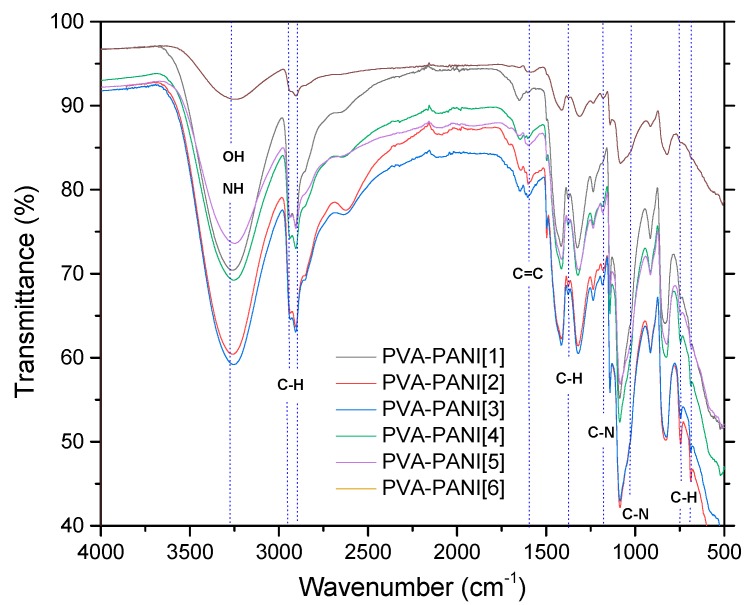
FTIR spectra of PVA–PANI gels at different PANI concentrations.

**Figure 5 molecules-24-00063-f005:**
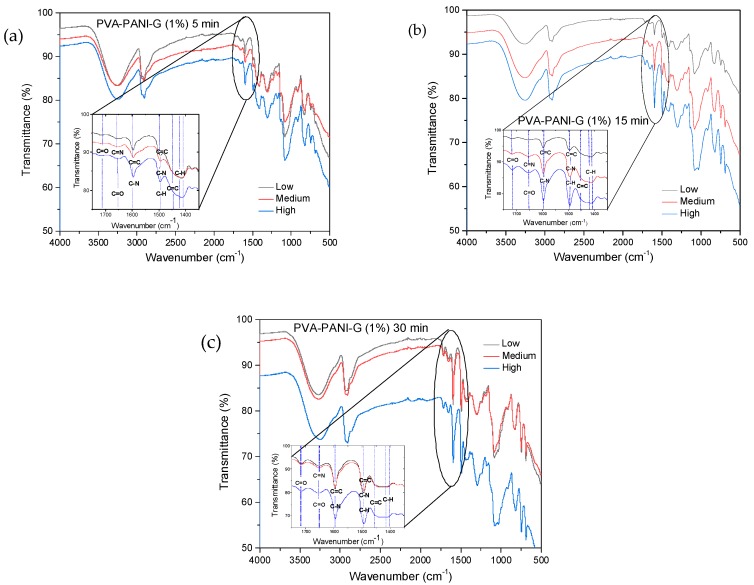
FTIR spectra of PVA–PANI gels at low, medium and high concentration of PANI activated with GA at 1% for: (**a**) 5 min, (**b**) 15 min, and (**c**) 30 min.

**Figure 6 molecules-24-00063-f006:**
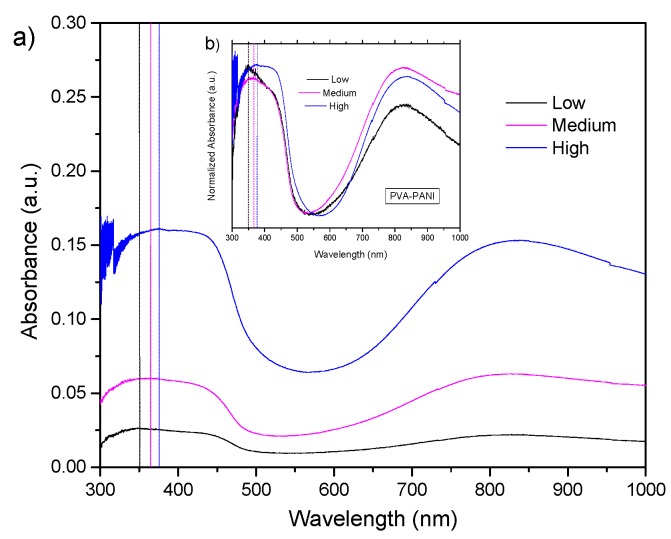
(**a**) UV–Vis-NIR absorbance and (**b**) normalized UV–Vis-NIR absorbance spectra of PVA–PANI thin films at low, medium, and high concentrations of PANI.

**Figure 7 molecules-24-00063-f007:**
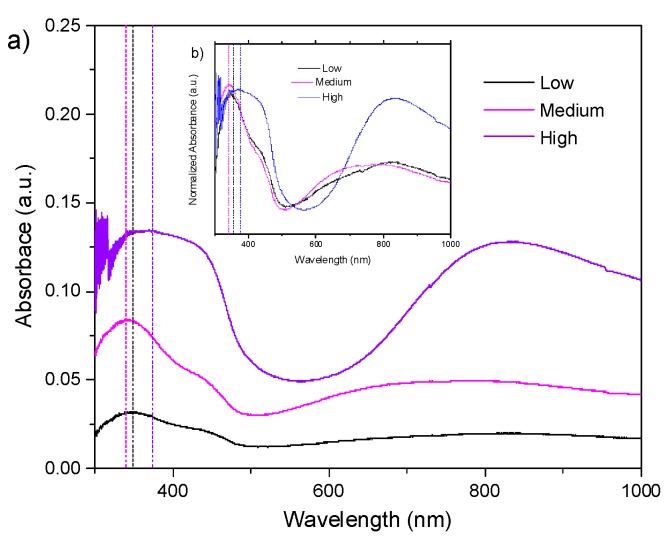
(**a**) UV–Vis-NIR absorbance spectra and (**b**) normalized UV–Vis-NIR absorbance spectra of PVA–PANI thin films activated with 1% GA for 30 min.

**Figure 8 molecules-24-00063-f008:**
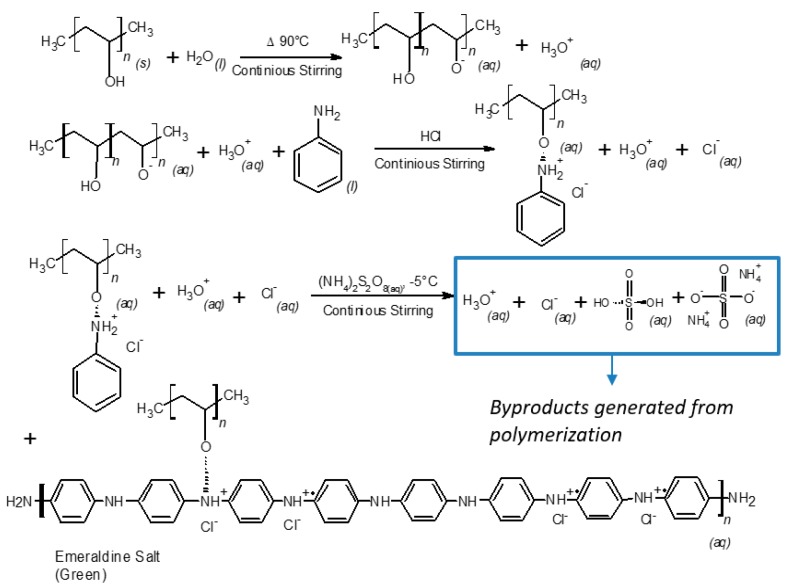
Synthesis route of PVA–PANI and generated byproducts.

**Figure 9 molecules-24-00063-f009:**
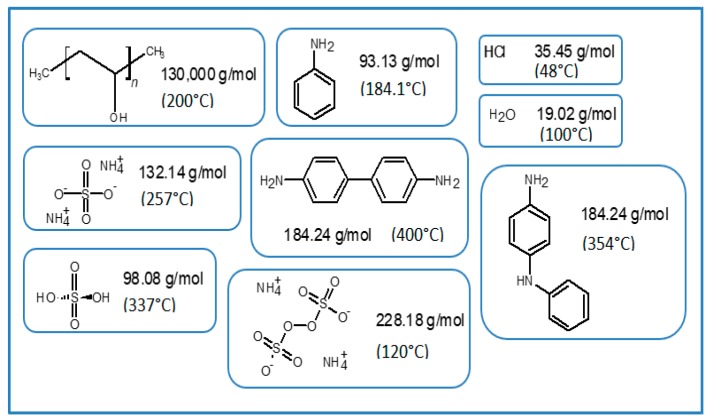
Molecular weight and transition temperatures of byproducts, sideproducts, and reagents present during synthesis.

**Figure 10 molecules-24-00063-f010:**
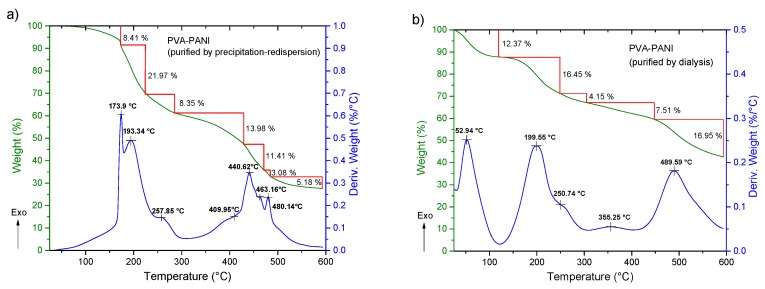
TGA-DTG of PVA–PANI purified by (**a**) precipitation–redispersion and (**b**) dialysis.

**Figure 11 molecules-24-00063-f011:**
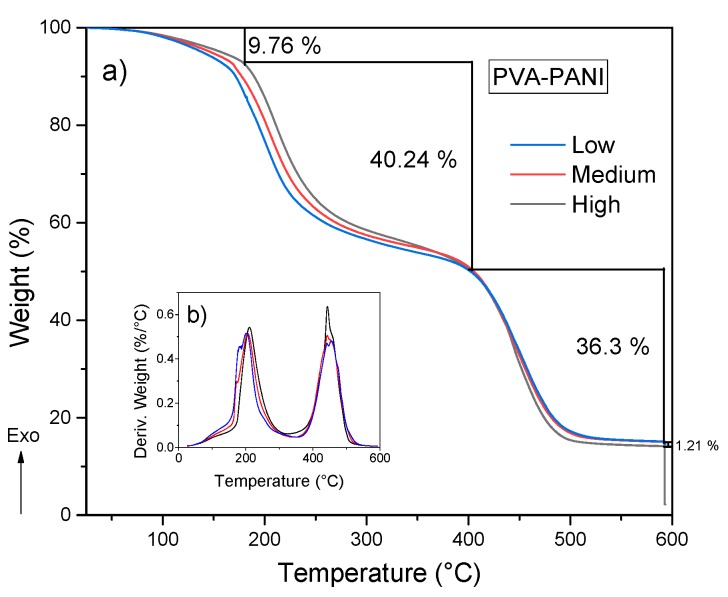
(**a**) TGA and (**b**) DTG of purified PVA–PANI gels at low, medium, and high concentrations of PANI.

**Figure 12 molecules-24-00063-f012:**
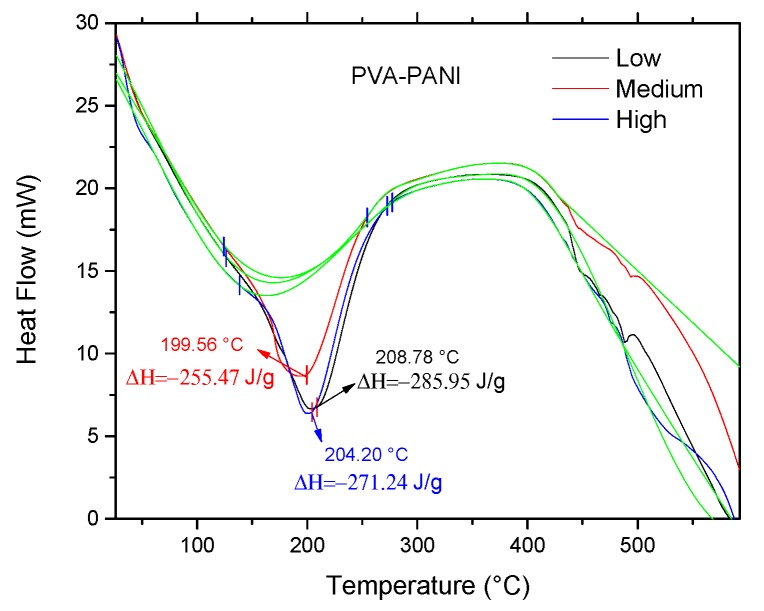
DSC of purified PVA–PANI gels at low, medium, and high concentrations of PANI.

**Figure 13 molecules-24-00063-f013:**
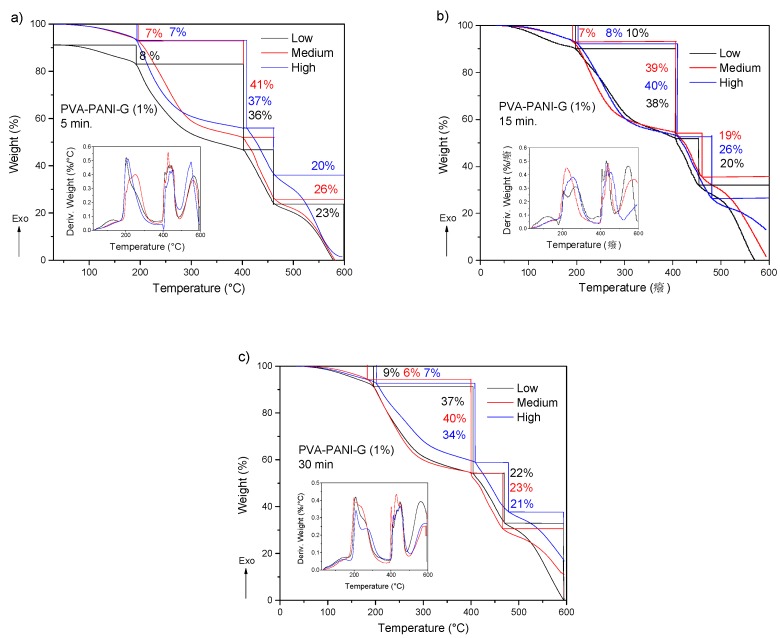
TGA and DTG thermograms of purified PVA–PANI-G gels, activated with 1% GA for (**a**) 5 min, (**b**) 15 min, and (**c**) 30 min.

**Figure 14 molecules-24-00063-f014:**
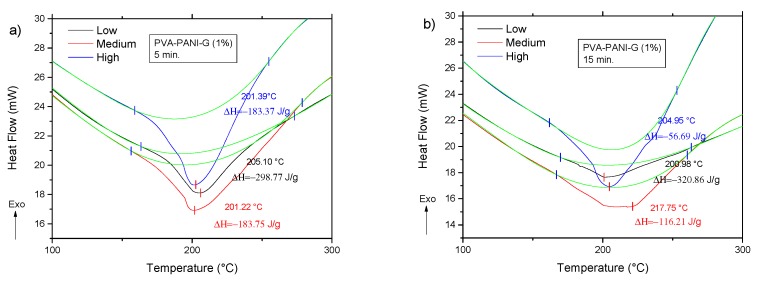
DSC thermograms of purified PVA–PANI-G gels, activated with 1% GA for (**a**) 5 min, (**b**) 15 min, and (**c**) 30 min.

**Figure 15 molecules-24-00063-f015:**
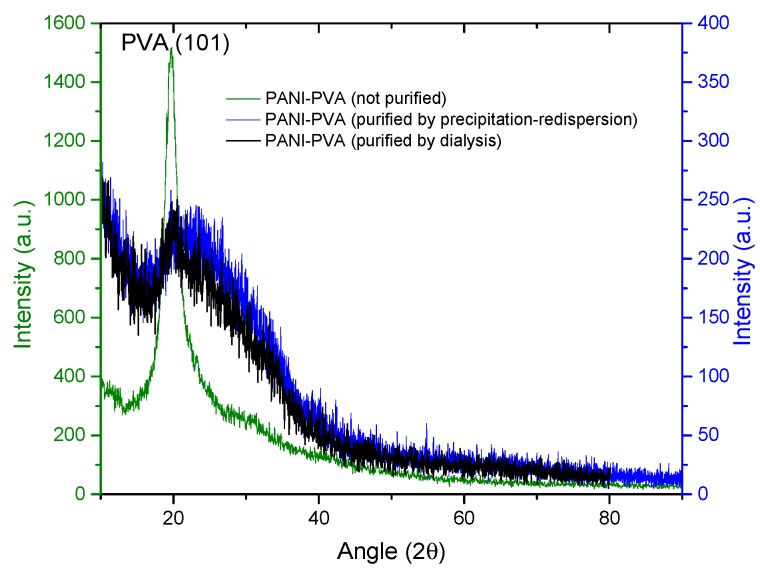
XRD diffractogram comparison for unpurified and purified PVA–PANI by precipitation–redispersion and dialysis.

**Figure 16 molecules-24-00063-f016:**
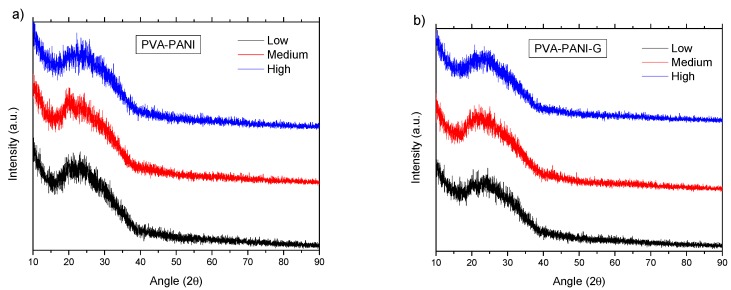
XRD diffractogram of (**a**) PVA–PANI thin films at different concentrations of aniline and (**b**) its comparison with PVA–PANI-G thin films.

**Figure 17 molecules-24-00063-f017:**
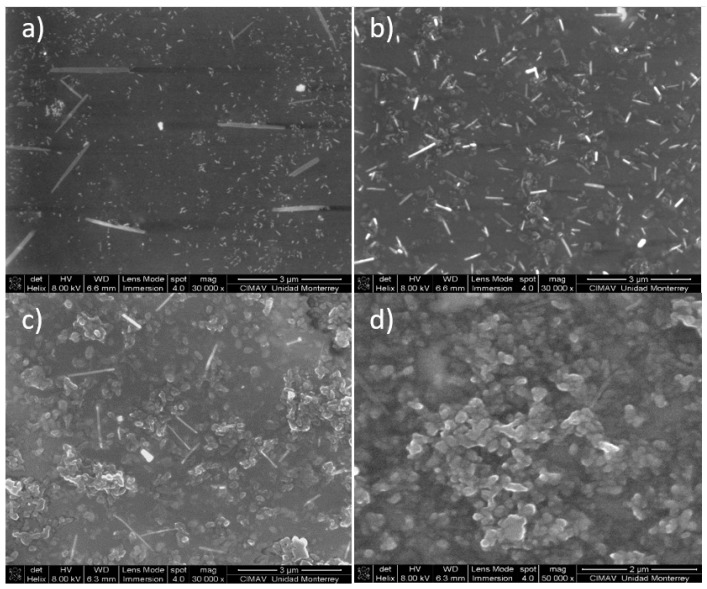
SEM micrographs of purified PVA–PANI thin films and its morphologic variations at (**a**) low concentration, (**b**) medium concentration, and (**c**,**d**) high concentration.

**Figure 18 molecules-24-00063-f018:**
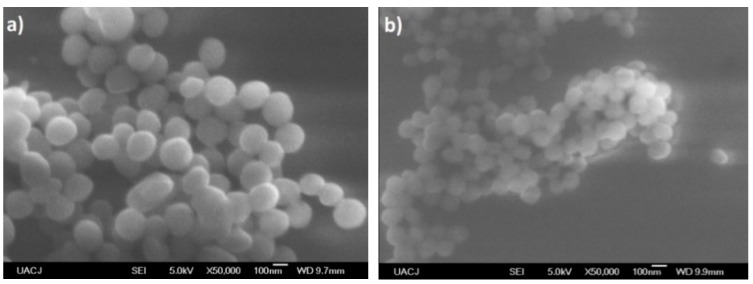
SEM micrographs of (**a**) PVA–PANI and (**b**) PVA–PANI-G purified by dialysis.

**Table 1 molecules-24-00063-t001:** Experiment of aniline concentration variation against PVA 5 wt %.

Concentration	Sample	PVA 5 wt %	Aniline	PVA–PANI (*wt*/*wt*)
**Low**	PVA–PANI-1	0.0202 mmol	3.5203 mmol	12.5 wt %
	PVA–PANI-2	0.0202 mmol	7.0640 mmol	25 wt %
**Medium**	PVA–PANI-3	0.0202 mmol	14.1281 mmol	37.5 wt %
	PVA–PANI-4	0.0202 mmol	17.6601 mmol	50 wt %
	PVA–PANI-5	0.0202 mmol	21.1921 mmol	62.5 wt %
**High**	PVA–PANI-6	0.0202 mmol	24.7241 mmol	75 wt %

**Table 2 molecules-24-00063-t002:** Monomer and intermediary molar ratios.

PVA (mmol)	Aniline (mmol)	Aniline (M)	HCl (mmol)	APS (mmol)	APS (M)
0.0202	3.532	0.070	100	3.532	0.070
0.0202	7.064	0.141	100	7.064	0.141
0.0202	10.596	0.212	100	10.596	0.212
0.0202	14.128	0.282	100	14.128	0.282
0.0202	17.660	0.353	100	17.660	0.353
0.0202	21.192	0.423	100	21.192	0.423

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
