# Peer review of "Purification and Glutaraldehyde Activation Study on HCl-Doped PVA–PANI Copolymers with Different Aniline Concentrations"

_molecules, 2018, doi:10.3390/molecules24010063_

Reviewer 1 Report

This manuscript deals with the synthesis and purification of polyvinyl alcohol-polyaniline composite with different concentrations of aniline monomers. A number of spectroscopic analyses, thermal analyses, and morphological studies have been carried out. The authors insist that the amounts of aniline have affected both purity and dispersion of the resulting products. Furthermore,  this material was treated with glutaraldehyde to enhance the potential material as active membrane for detecting protein or enzyme. Despite hard working, this paper cannot be accepted without appropriate revisions. My comments are as follows:

1. According to Figure 1, the scheme indicates that the chemical bond between amine nitrogen group of PANI and hydroxyl group of PVA, resulting in the formation of ether crosslink. According to the basic of polymer chemistry, the term blending refers to the physical and mechanical mixing, while the copolymerization includes a number of chemical bonds between more than two polymers. Considering these, the material prepared in this work seems to be copolymer rather than blended composite.

2. It is well known fact that the different concentrations of aniline not only cause different degree of polymerization but also result in different charge transport properties and different chain length of polymer. Thus, it is reasonable to predict that the sensitivity toward glutaraldehyde will be dependent on the crystallinity and doping level of PVA-PANI.

3. According to previous reports on PVA-PANI, water-dispersible steric stabilizers including PVA, PAA, and PVP also act as tailoring agent that directly affect the morphology of PANI nanomaterials. The authors should add explanation in the manuscript.

4. In Figure 13, I cannot see the exact difference between XRD patterns from different samples, thereby you have to present each pattern separately.

5. There are a number of work on the highly conductive PANI nanomaterials. Considering synthetic conditions represented in the manuscript, I don't think this work is distinguishable from previous work on PANI. The authors have to conduct further investigation on the electrical properties of PVA-PANI with different aniline concentrations, such as electrical conductivity measured from 4-probe method, sheet resistivity of the film.

6. There are very few mentions regarding the glutaraldehyde. The authors are recommended to adjust the amounts of glutaraldehyde to identify the optimum concentration of glutaraldehyde for the active membrane.

Judging from these facts, this acceptance of this manuscript can be considered only after appropriate revisions. Otherwise, the editorial office can reject this article if the revision is not satisfactory.

Author Response

Dear Reviewers:

We sincerely thank the reviewers for the time invested in reviewing our paper as well as their valuable comments. We have revised the paper taking into account the comments provided. We are now submitting the revised manuscript for review and potential publication.

Response to reviewer #1

General comments

Point 1) According to Figure 1, the scheme indicates that the chemical bond between amine nitrogen group of PANI and hydroxyl group of PVA, resulting in the formation of ether crosslink. According to the basic of polymer chemistry, the term blending refers to the physical and mechanical mixing, while the copolymerization includes a number of chemical bonds between more than two polymers. Considering these, the material prepared in this work seems to be copolymer rather than blended composite.Author response: After analyzing references and the obtained results for the determination of molecular structure of the material the authors agree, and modifications where made in the text.

Point 2) It is well known fact that the different concentrations of aniline not only cause different degree of polymerization but also result in different charge transport properties and different chain length of polymer. Thus, it is reasonable to predict that the sensitivity toward glutaraldehyde will be dependent on the crystallinity and doping level of PVA-PANI.

Author response: We agree that variations of molecular weight with an increment of monomer are expected to take place, as discussed on the variation of benzenoid and quinoid sections of PANI by UV-Vis characterization. Since the synthesized material tends to be amorphous as shown by the XRD results the sensitivity of glutaraldehyde were analyzed by DSC characterization for low, medium and high concentration at 5, 15 and 30 minutes of activation.

Point 3) According to previous reports on PVA-PANI, water-dispersible steric stabilizers including PVA, PAA, and PVP also act as tailoring agent that directly affect the morphology of PANI nanomaterials. The authors should add explanation in the manuscript.

Author response: Discussion has been adjusted for SEM characterization focusing on the stabilizing effect of PANI and the molecular order of PANI in the dispersion.

Point 4)  In Figure 13, I cannot see the exact difference between XRD patterns from different samples, thereby you have to present each pattern separately.

Author response: The format for Figure 13 was modified taking into account the reviewer’s comment.

Point 5) There are a number of works on the highly conductive PANI nanomaterials. Considering synthetic conditions represented in the manuscript, I don't think this work is distinguishable from previous work on PANI. The authors have to conduct further investigation on the electrical properties of PVA-PANI with different aniline concentrations, such as electrical conductivity measured from 4-probe method, sheet resistivity of the film.

Author response: Semiconductivity of the purified PVA-PANI films was further analyzed through the change in bandgap due to absorbance shifting at different concentrations and activation times with gluteraldehyde. The corresponding Figure was added to the supplementary material.

Point 6) There are very few mentions regarding the glutaraldehyde. The authors are recommended to adjust the amounts of glutaraldehyde to identify the optimum concentration of glutaraldehyde for the active membrane.

 Author response: Experiments of material activation with gluteraldehyde at a concentration of 2.5% where carried out and added as supplementary material. Obtained results for FTIR show further reduction of PANI.

Finally, about English language and style: We have reviewed the manuscript carefully and made changes were considered necessary. We believe the improved grammar and punctuation will better serve the reader.

Reviewer 2 Report

Aguirre et al. reported a systematic investigation on purification and glutaraldehyde process of HCl doped PVA-PANI blends with different aniline concentrations. Detailed experimental results were presented and the difference between processes were deeply discussed. However, some of the important characterizations of material itself is missing. I thus recommend this manuscript can be published in Molecules after a major revision. Followings are the remarks for the author: 

1. The authors designed PVA-PANI blends with different aniline concentrations. In the manuscript, only theoretical PVA/PANI ratios were given in the Table. What is the actual PVA/PANI ratios with different aniline concentrations? The authors should extract the ratios from NMR or element analysis.

2. The NMR of different PVA-PANI blends conditions should be provided, which can understand the changing of PVA/PANI ratio.

3. In Figure 2, what are the sharp peaks at 2 and 3.25ppm?

4. In Figures 6 and 7, the peak position change is hard to see. A normalization spectrum should be provided.

5. By Figure 14 and 15, a clear morphology difference can be observed with different purification process. How is the XRD results of the PVA-PANI blends purified by different method? A significant difference is expected due to morphology changes.

Author Response

Dear Reviewers:

We sincerely thank the reviewers for the time invested in reviewing our paper as well as their valuable comments. We have revised the paper taking into account the comments provided. We are now submitting the revised manuscript for review and potential publication.

Response to reviewer #2

General comments

Point 1. The authors designed PVA-PANI blends with different aniline concentrations. In the manuscript, only theoretical PVA/PANI ratios were given in the Table. What is the actual PVA/PANI ratios with different aniline concentrations? The authors should extract the ratios from NMR or element analysis.

 Author response: We agree that theorical PVA/PANI ratios may vary due to different degrees of polymerization and molecular weight of PANI at different aniline concentrations. Actual PVA-PANI ratio for the presented 1H NMR spectra was extracted and added to the manuscript.

Point 2. The NMR of different PVA-PANI blends conditions should be provided, which can understand the changing of PVA/PANI ratio.

 Author response: 1H NMR characterization was used only for molecular structure confirmation and further analysis for different aniline concentrations were carried out by FTIR in order to obtain a qualitative relation between each concentration. From the obtained results a low medium and high concentration were selected for further characterization studies.

Point 3. In Figure 2, what are the sharp peaks at 2 and 3.25ppm?

 Author response: Discussion for all prominent 1H NMR peaks was added to the manuscript.

Point 4. In Figures 6 and 7, the peak position change is hard to see. A normalization spectrum should be provided.

 Author response: Normalization and intensity for the absorption bands are now provided as an inset for both Figures 6 and 7 in order to correlate band shifting.

Point 5. By Figure 14 and 15, a clear morphology difference can be observed with different purification process. How is the XRD results of the PVA-PANI blends purified by different method? A significant difference is expected due to morphology changes.

 Author response: Discussion about the relation between XRD and SEM results was added to the manuscript, taking into account that nucleation sites for PVA-PANI purified by precipitation-redispersion and dialysis vary due to the dispersion medium.

Finally, about English language and style: We have reviewed the manuscript carefully and made changes were considered necessary. We believe the improved grammar and punctuation will better serve the reader.

Round  2

Reviewer 1 Report

I recommend this paper to be published in Molecules journal.

Reviewer 2 Report

The authors have addressed all comments form the reviewer. Now this manuscript can be published in Molecules